# Positive or Negative: The Heterogeneities in the Effects of Urban Regeneration on Surrounding Economic Vitality—From the Perspective of Housing Price

**Meng Yuan** and **Hongjuan Wu** *

School of Management Science and Real Estate, Chongqing University, Chongqing 400044, China; yuanm@cqu.edu.cn
* Correspondence: hongjuanwu@cqu.edu.cn

**Abstract:** Urban regeneration is a sound, sustainable urban development strategy globally. In China, promoting urban regeneration has become the national sustainable urban strategy. Under the resource constraint, it is necessary to understand what benefits different types of urban regeneration projects can contribute to urban development. Much research has contributed to evaluating the benefits of urban regeneration on the project scale. The systematic investigation of their effects on surrounding economic vitality, especially the comparison between different types of projects, is relatively downplayed. This paper aims to evaluate and compare the effects of urban regeneration projects on surrounding economic vitality by calculating the change in housing prices. Chongqing is selected as the case city. Based on the housing transaction data from 2015 to 2021, a staggered difference-in-difference method is employed to capture the results. The finding shows that the overall effects are significantly negative, especially within a 400–800 m radius. Higher investment or better project location relates to stronger negative effects. Moreover, the differences in effects among the three types of urban regeneration projects are clearly revealed from diverse perspectives. It provides a valuable reference for policymakers and urban planners to make urban regeneration planning better by considering comprehensive benefits.

**Keywords:** economic vitality; urban regeneration; housing price; difference-in-difference method; China

## 1. Introduction

With the rapid increase in urbanization, a massive population has migrated and agglomerated into the urban area. According to the report of UN-Habitat [1], the world's urbanization rate will climb to around 68% by 2050. Urbanization brings about tremendous benefits for human society, but problems have also occurred, such as low intensive land use, social inequality, and ecosystem degradation [2]. Under this context, sustainable urban development has gradually become an urgent need for cities worldwide. It aims to meet the demands of both current generation and future development requirements through sustainable urban growth and structural evolution [3]. Today, urban regeneration has become a critical approach to improving urban areas by rehabilitating dilapidated buildings, revitalizing decayed areas, and redeveloping blighted spaces [4]. It plays a significant role in sustainable urban development strategies globally [5]. For example, the First Berlin Renewal Program in Germany supported private investment in housing stock renovation to improve the condition and value of properties in decayed areas after World War II [6]. The Neighborhoods-in-Bloom Urban Revitalization Program in Richmond was released to revitalize decayed areas without sufficient public investment [7]. Thus, promoting urban regeneration has long been a global topic.

The urban regeneration practices in China are representative and unique, which provides a sample for research on urban-related issues worldwide. In 2015, the Cen-

tral Work Conference of Urbanization proposed a new goal of sustainable urban regeneration in China, representing that urban regeneration's connotation was shifted from large-scale demolition and reconstruction to comprehensive approaches [8]. In 2020, the 14th 5-year plan emphasized that promoting urban regeneration on a sustainable path is the national urban strategy [9]. Today, based on the physical change level and cultural values, there are three main approaches to urban regeneration: rehabilitation, revitalization and redevelopment [10]. Rehabilitation involves modest physical enhancements at the neighborhood scale, focusing on maintaining and improving existing structures, including elevator installation, building structure reinforcement, purification of the neighborhood environment, and provision of facilities and amenities, etc. [11]. Revitalization fosters urban vitality and heritage conservation by repurposing historically significant but underutilized structures, such as industrial buildings or culturally valuable neighborhoods [12]. Meanwhile, redevelopment entails large-scale demolition and reconstruction of deteriorated buildings [13] or polluted brownfields and greyfields [14].

In most cases, enhancing sustainability always means reducing market profitability [15]. In China, the government sectors are responsible for investing and initiating urban regeneration in the early stages [16]. The reduction in large-scale demolition and reconstruction means there are fewer investment opportunities for market power to participate in urban regeneration. Therefore, public resources are the primary support for the project promotion. Yet, due to resource constraints, it is impossible to conduct a large number of urban regeneration projects in the short term. So, evaluating what benefits different types of urban regeneration projects can contribute to the city is of great necessity for policymakers and urban planners to make better choices [17].

Plenty of research studies currently contribute to the performance assessment of urban regeneration projects [18,19]. Yet, most of these studies focus narrowly on project-specific scopes and fail to elucidate the broader impacts on surrounding areas. Some scholars realized this in their studies, whereas they only focus on one specific type of building/neighborhood, such as industrial buildings, shanty towns, and historical sites in different backgrounds [20–22]. That is to say, there remains no scholarly consensus on the correlation between external effects and intrinsic features of urban regeneration. To address the research gap, this paper aims to evaluate the effects of urban regeneration on the surrounding area and compare the variances across different types of projects in China. Economic vitality is essential for sustainable urban development [23], and stimulating economic vitality is a recognized crucial benefit of urban regeneration initiatives [24]. Therefore, taking the effects on economic vitality of urban regeneration as the main research objective is necessary in practice and academia. This study employs housing price as the indicator of economic vitality, answering two main questions: Whether and to what extent urban regeneration influences nearby economic vitality? How does urban regeneration with multiple characteristics exert different effects? Chongqing is selected as the case city. The results can provide references for policymakers and urban planners to make better decisions on urban regeneration.

The structure of this paper is arranged as follows: Section 2 reviews the literature about the characteristics and the effects of urban regeneration projects on economic vitality. In Section 3, the study area is described, and the data collection is explained. It also illustrates how the difference-in-difference method is used to investigate the heterogeneity and how additional tests are conducted to improve the robustness of the results. Section 4 reports the benchmark regression results and the effects of projects with different characteristics. In Section 5, the corresponding discussion is provided for explaining the results and comparing the differences. The final section summarizes the whole paper and makes recommendations for future research.

## 2. Literature Review

### 2.1. The Effects of Urban Regeneration on Economic Vitality in Different Scope

Plenty of research studies exist on economical evaluation of urban regeneration from multiple dimensions. The effects of regenerated areas are widely evaluated by establishing an assessment system framework based on sustainable development theory [25–27]. A comprehensive assessment composed of several indicators has generally been conducted to evaluate economic sustainability, including employment rate, tax revenue, resident income, benefit–cost ratios, and so on [24,28–30]. Moreover, Wang, et al. [31] employed a data envelope analysis (DEA) method to estimate the benefits by taking investment and financial expenditure as input indicators. Della Spina, et al. [32] measured economic sustainability and financial feasibility by maximizing the achievement based on balancing stakeholders' interests. In addition to the areas directly affected by urban regeneration, some research studies also contribute to evaluating economic effects on a broader scale. Liu, et al. [33] explored the economic benefits of the urban regeneration industry and forecast its contribution to GDP by 2030. Albanese, et al. [34] found that urban regeneration has done little work to stimulate the growth of the local economy in the short to medium term.

The linkage between real estate dynamics and economic vitality is well-established in academia [35,36]. Scholars have demonstrated that urban regeneration projects, especially redevelopment-type projects, can significantly alter the real estate market. These changes extend beyond the projects' confines but also affect the properties close to them [37,38]. Therefore, the impact on neighborhood property values constitutes a critical area in the field of urban regeneration's economic effects evaluation [39–41]. Some scholars investigated whether urban regeneration had an influence on retailing or rents in the surrounding areas [42–45]. Additionally, housing prices serve as a representative indicator reflecting economic development [46] due to their direct correlation with residents' willingness to pay for better living conditions after regeneration [47]. With this in mind, housing prices are considered suitable for this study to capture the change in economic vitality sensitively and accurately.

### 2.2. The Correlation between Urban Regeneration Characteristics and Economic Effects

Evaluating the effects on economic vitality is of great significance for the performance assessment of urban regeneration. Hall [48] proposed an "outward-looking" approach to assess the regeneration policy, which broke the limitation of area-based initiatives (ABIs) and explored the effects in broader urban areas. Based on this, Saiu [49] further developed a conceptual framework of Outwards Regeneration Effects (ORES), indicating that it differs for ORE on project-specific features. Other research studies have also demonstrated that different project features, such as types, investments, and locations, closely correlate with the economic effects of urban regeneration.

Evaluating urban regeneration's effects from the perspective of specific project types is a central theme in most studies. Industrial sites [21], heritage buildings [12], and squatter settlements [50] are all found to exert influence on surrounding housing prices. While these studies provide invaluable insights, most have been limited to examining specific types of projects, leaving cross-type comparisons largely underexplored. This oversight has resulted in a subdued understanding of the correlation between different types of regeneration projects and their economic effects. Recognizing this gap, some scholars have initiated cross-type studies, classifying projects by various criteria, such as connection to existing sites [14], applicant groups [51], land use [52], and so on. Compared to Western and other developed regions, urban regeneration projects are officially classified by renewal approach in some cities of China, including rehabilitation, revitalization, and redevelopment [53]. Nevertheless, the regeneration approach affects housing prices heterogeneously and has been infrequently explored in existing studies [54]. Therefore, comparing three types of projects has the potential to fill the research gap of insufficient attention to the correlation between the regeneration approach and the economic effects of urban regeneration.

Many research studies have stated that the effects of the distance change to the projects are heterogeneous. The renewal of urban villages exerts different economic influences within a radius of 1000 m and 2000 m [47]. The value of properties near Tong Lau, a type of old multi-owned house, increased by 5.63% within a radius of 100 m and 7.35% within a radius of 200 m [55]. The preservation of heritage or landmarks has been found to lead to a price premium of nearby property by 1.7%, 1.4%, and 0.5% within a radius of 50 m, 100 m, and 200 m [12]. Liu and Liu [56] plot the irregular curve of average property prices close to the heritage adaptive reuse projects.

The effects on economic vitality are also related to the investment in projects. Every euro invested in urban renewal can increase the property value by 0.06 to 1.35 euros in the First Berlin Renewal Program [6]. In the Neighborhood-in-Bloom (NiB) program, every dollar may translate into $2 to $6 in incremental land value [7]. Koster and Rouwendal [57] revealed that the prices of surrounding buildings increased by 1.5–3%, with the increase in investment for historical heritage per 1 million square kilometers. However, the small-scale investment may not strongly influence the nearby housing price [58].

The effects of urban regeneration on economic vitality are also affected by project location. Waltl [59] divided Sydney into 16 regions and 3 clusters according to price segments and geography and found that the suburban had the highest appreciation rates opposite to that of the inner city. Van Duijn, Rouwendal and Boersema [39] demonstrated that the positive effects on housing prices by redeveloping industrial heritage would disappear when the projects in the largest cities were excluded. Diamond and McQuade [60] also found that the housing price in low-income neighborhoods was increased by the Low-Income Housing Tax CREDIT (LIHTC) but decreased by 2.5% in higher-income areas. Moreover, some studies have proved that urban regeneration plays different economic roles between high-price and low-price regions [61,62]. To explore the influence of location, some research studies adopted specific methods to quantify the location value [63]. The most widely applied quantification method is to measure the accessibility of public facilities [64,65].

## 3. Methodology

### 3.1. Study Area

Chongqing is located in southwest China, and is one of the largest Chinese cities, with a population of 32 million and an area of 82,400 km$^2$. As a pioneer of urban development in China, Chongqing has conducted a large number of urban regeneration practices, which provides plenty of cases and resources for study from various perspectives [66]. By 2021, 33.75 million m2 of old neighborhoods and 27.88 million m2 of shanty towns have been regenerated. Many occurred in the central district area consisting of nine districts, namely, Yuzhong, Dadukou, Jiangbei, Shapingba, Jiulongpo, Nan'an, Beibei, Yubei, and Banan. In 2018, the municipal pilot urban regeneration projects were implemented in four of the above nine districts (Shapingba, Jiulongpo, Yubei, and Nanan). In 2021, the national government awarded two of the above nine districts (Yuzhong and Jiulongpo) as pilot cities (districts) for promoting urban regeneration. Therefore, this paper chooses the central district area of Chongqing as the study area.

### 3.2. Data

3.2.1. Information of Urban Regeneration Projects

This paper selects 37 urban regeneration projects completed after 2015. The selected projects represent all three types of regeneration approaches, different investment scales, and diverse project locations. The geographical distribution of selected urban regeneration projects is shown in Figure 1. Their information is collected through field investigation and collection of government documents.

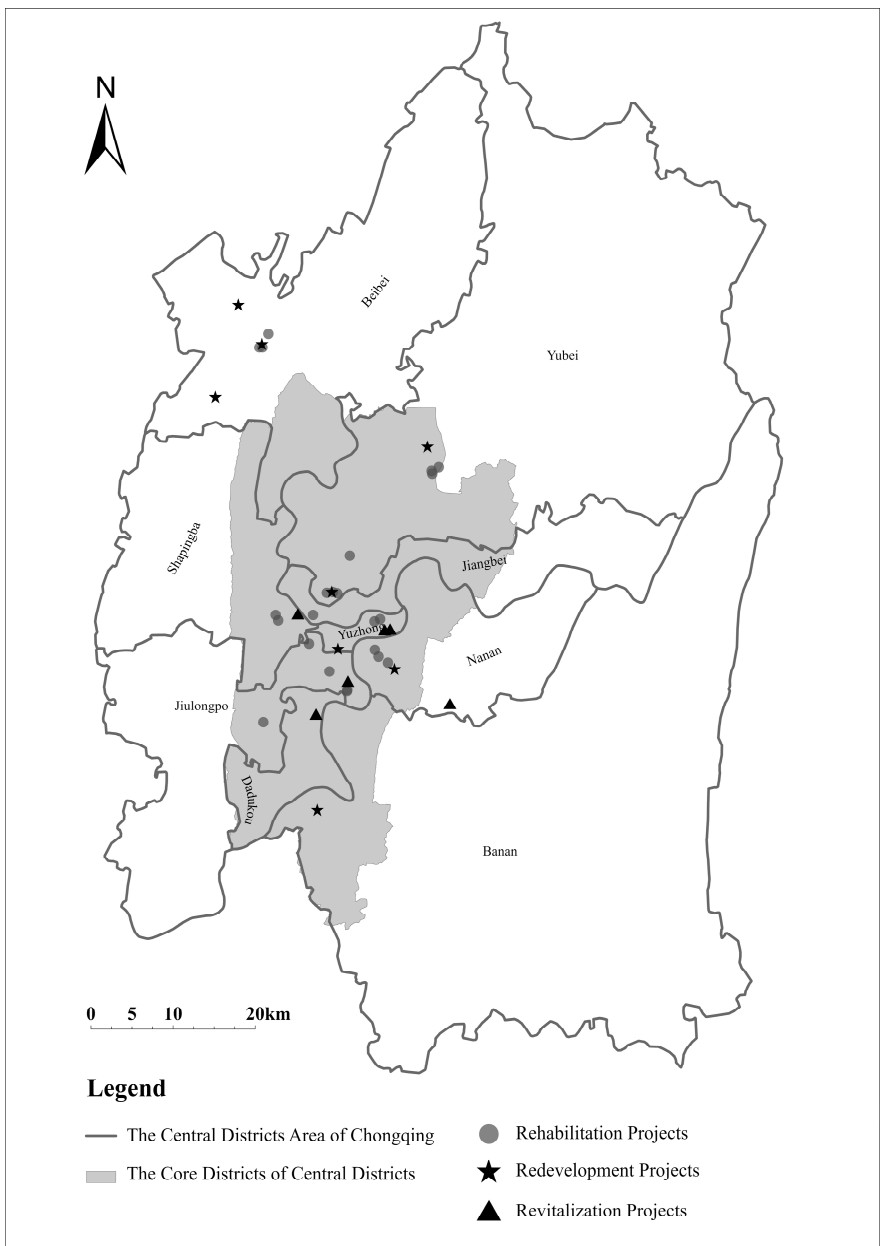

**Figure 1.** Case study area: The main districts area in Chongqing.

According to previous studies, location condition, which is generally characterized by distance to public facilities, is one of the significant factors affecting the effects of urban regeneration [67]. Seven specified facility types are chosen, including transportation, administration, commerce, education, health care, culture and sports, finance and post. Their location is represented by the interest points from the Baidu Map. Referring to Li, et al. [68], this paper employs Shannon information entropy to determine the weight value of each public facility (See Table A1). Since TOPSIS is a widely used quantifying method characterized by low data requirement, simple calculation processing, and ease of operation, it is chosen to measure the location value of urban regeneration projects by utilizing the weight value above mentioned. The higher location value means that the project is in a better location. The results are shown in Table A2.

### 3.2.2. Information of Housing Transactions

Housing transaction information is collected via web scraping methods using a Python script from one of the largest real estate portals, Lianjia. After eliminating duplicates and

outliers, there were 113,544 housing transactions from January 2015 to December 2021. A neighborhood has lots of transaction data with different characteristics simultaneously. The changes in average transaction prices in the same neighborhood before and after the implementation of projects can be regarded as the effects of urban regeneration on surrounding economic vitality. The housing transaction is regarded as only affected by the nearest urban regeneration project. Therefore, the data regarding the neighborhood located in the buffer zone of 1000 m of two different types of projects are also excluded to distinguish the effects of projects with varying approaches of regeneration. This paper takes half a year as a base period to build panel data. Finally, 21,316 groups of data about average housing prices are obtained.

According to the hedonic model of housing price, the collected data also includes the characteristics of transacted properties from the dimensions of physical structure, location, and neighborhood. The physical structure consists of the floor area, the age of the buildings, the number of bedrooms, etc. In addition to the distance to the closest urban regeneration project, according to Lv, et al. [69], the locational characteristics also include the distances to the nearest subway station, college, and commercial area. The public facilities' point of interest (POI) is obtained from the Baidu Map, and the distance is measured through the geographic information system (GIS). The neighborhood characteristics are spatial correlation attributes reflecting similar prices, structure characteristics, environment, etc., in the cluster of houses usually presented as gated communities in the context of China [70]. Gated communities' building density and living environment quantitatively exert significant influence on housing prices [71,72]. Consequently, the plot ratio and the greening rate are utilized to measure neighborhood characteristics. These characteristics are used as the control variables in the DID model. Their definition and descriptive statistics are shown in Table 1. The complete information of all urban regeneration projects is shown in Table 2.

**Table 1.** The definition and descriptive statistics of variables.

| Variables | | Definition | Mean | Sd | Min | Max |
|---|---|---|---|---|---|---|
| **Dependent Variable** | | | | | | |
| | Price | The average price of house (1 thousand yuan/m$^2$) | 11.290 | 3.585 | 1.638 | 38.690 |
| **Independent Variable** | | | | | | |
| | Area | The floor area of house (m$^2$) | 93.820 | 51.00 | 16.050 | 4324.000 |
| Physical Structure | Dec | Whether it is decorated: Undecorated (0), simple-decorated (1), well-decorated (2) | 1.350 | 0.629 | 0.000 | 2.000 |
| | Bed | The number of bedrooms | 2.442 | 0.831 | 1.000 | 14.000 |
| | Ele | Whether it has elevator: no (0), yes (1) | 0.882 | 0.311 | 0.000 | 1.000 |
| | Age | The number of years between built and sale | 3.427 | 4.989 | 0.000 | 57.000 |
| Neighborhood | Pr | The plot ratio of the community | 3.277 | 1.608 | 0.020 | 10.000 |
| | Gr | The greening rate of the community | 0.328 | 0.073 | 0.100 | 0.700 |
| Location | Dissub | The distance to nearest subway station (km) | 0.712 | 0.706 | 0.029 | 23.220 |
| | Dise | The distance to nearest college (km) | 2.661 | 2.026 | 0.044 | 12.460 |
| | Disc | The distance to nearest commercial area (km) | 0.614 | 0.837 | 0.000 | 22.280 |
| | Dispro | The distance to nearest regeneration project (km) | 2.723 | 3.034 | 0.000 | 21.410 |
| Projects | Method | The regeneration method of projects: rehabilitation (0), revitalization (1), redevelopment (2) | 0.479 | 0.758 | 0.000 | 2.000 |
| | Investment | The investment on projects (one hundred million) | 2.199 | 4.416 | 0.001 | 20.000 |
| | Location | The location value of the projects | 0.416 | 0.139 | 0.048 | 0.695 |

**Table 2.** The attributes of urban regeneration projects.

| Project | Period | Investment (One Hundred Million) | Regeneration Approach | Location Value |
|---|---|---|---|---|
| LJC | 2019.4–2019.7 | 0.030 | Rehabilitation | 0.083 |
| TSGC | 2017.2–2017.9 | 0.076 | Rehabilitation | 0.160 |
| NRC | 2017.8–2019.11 | 1.300 | Rehabilitation | 0.532 |
| CSC | 2018.10–2019.4 | 0.029 | Rehabilitation | 0.162 |
| QC | 2020.4–2020.8 | 0.060 | Rehabilitation | 0.456 |
| TFYC | 2020.9–2021.2 | 0.053 | Rehabilitation | 0.077 |
| TFJC | 2020.9–2021.2 | 0.081 | Rehabilitation | 0.122 |
| TTJC | 2020.9–2021.2 | 0.219 | Rehabilitation | 0.112 |
| RC | 2019.1–2019.12 | 0.014 | Rehabilitation | 0.355 |
| FRC | 2019.1–2019.12 | 0.010 | Rehabilitation | 0.323 |
| LC | 2019.1–2019.12 | 0.002 | Rehabilitation | 0.308 |
| SC | 2019.1–2019.12 | 0.005 | Rehabilitation | 0.074 |
| JC | 2019.1–2019.12 | 0.003 | Rehabilitation | 0.083 |
| KNC | 2019.1–2019.12 | 0.001 | Rehabilitation | 0.097 |
| KC | 2020.1–2020.11 | 0.870 | Rehabilitation | 0.171 |
| ZLC | 2020.10–2021.3 | 0.096 | Rehabilitation | 0.180 |
| SRC | 2020.3–2020.6 | 0.182 | Rehabilitation | 0.220 |
| DC | 2018.3–2019.12 | 0.120 | Rehabilitation | 0.071 |
| KLC | 2019.1–2019.12 | 0.078 | Rehabilitation | 0.065 |
| XQRC | 2019.3–2020.1 | 1.300 | Rehabilitation | 0.281 |
| WC | 2020.1–2020.6 | 0.020 | Rehabilitation | 0.467 |
| YC | 2020.10–2021.1 | 0.030 | Rehabilitation | 0.467 |
| CVS | 2019.9–2019.12 | 0.120 | Rehabilitation | 0.229 |
| FDSS | 2019.6–2020.12 | 0.046 | Redevelopment | 0.077 |
| SHSS | 2020.1–2020.5 | 1.100 | Redevelopment | 0.695 |
| YSS | 2015.12–2017.12 | 8.884 | Redevelopment | 0.506 |
| YJSS | 2017.12–2019.12 | 0.060 | Redevelopment | 0.083 |
| SRSS | 2019.2–2021.9 | 12.000 | Redevelopment | 0.412 |
| SFSS | 2015.6–2021.1 | 6.300 | Redevelopment | 0.074 |
| LSS | 2017.7–2020.8 | 2.700 | Redevelopment | 0.466 |
| XMSS | 2019.12–2020.8 | 11.810 | Redevelopment | 0.149 |
| SHD | 2017.5–2021.9 | 20.000 | Revitalization | 0.283 |
| MCLHD | 2017.11–2019.9 | 1.200 | Revitalization | 0.263 |
| IMP | 2015.12–2016.12 | 1.700 | Revitalization | 0.048 |
| RTC | 2018.3–2020.3 | 0.300 | Revitalization | 0.476 |
| ICCIP | 2015.9–2016.10 | 5.000 | Revitalization | 0.188 |
| IM | 2015.1–2016.12 | 6.800 | Revitalization | 0.384 |

*3.3. The Difference-in-Difference Method*

The difference-in-difference method (DID) is widely used to measure the causal effects of a given intervention. The first difference is the change in outcomes before and after the intervention, while the second difference is the change between treatment groups versus control groups. The mathematical expression is as follows:

$$\left(Y_{treat}^{post} - Y_{treat}^{pre}\right) - \left(Y_{control}^{post} - Y_{control}^{pre}\right)$$

In this paper, the DID methods capture the relative change in nearby housing prices before and after implementing urban regeneration projects. Such relative change can be regarded as the estimation of economic effects because it excludes other potential influences. Since the start and end times of urban regeneration projects are different, this paper utilizes a staggered DID method, which allows the difference in the treatment period. Considering the hysteresis of economic effects, the panel data is constructed by taking half a year as the base period. It equals the coefficient on the interaction of a post-treatment dummy and a treatment dummy in the classical DID method. The hedonic price model (HPM) is generally utilized to measure the effects of housing-specific attributes, for the property

price is also determined by physical structure, neighborhood, and locational characteristics. Therefore, the final regression model in this paper is the combination of DID and HPM. The baseline model is as Equation (1).

$$P_{ijt} = \alpha + \beta_0 * DD_{ijt} + \lambda Z_{it} + v_i + u_t + \varepsilon_{it} \tag{1}$$

where $P_{ijt}$ indexes the average price of transaction housing $i$ at the time $t$, and the subscript $j$ indicates the urban regeneration project which is closest to housing $i$. The binary dummy variable $DD_{ijt}$ is the interaction of the post-treatment dummy, $Post_{ij}$ and the treatment dummy $Treat_{ij}$. This paper takes the transactions within 1000 m as the treatment group, because it is acceptable for residents to obtain public service in 15 min, nearly 1000 m according to the common walking speed. It also means that urban regeneration is considered to exert little effect on the property beyond 1000 m of projects. Therefore, $DD_{ijt} = 1$ if the housing $i$ is within 1000 m and transacted after the regeneration of the project $j$, otherwise, $DD_{ijt} = 0$. $\beta_0$ is the estimation of economic effects. $Z_{it}$ is a multiple vector of housing-specific physical, neighborhood, and locational characteristics. $u_t$ is the time-fixed effect reflecting the temporal trend of prices, while $v_i$ is the individual-fixed effect the unobserved time-varying factors of housing. $\varepsilon_{it}$ is the error term.

The benchmark regression of Equation (1) estimates the average treatment effects of urban regeneration on nearby property prices. This research further investigates whether the effects of urban regeneration on economic vitality are different in areas with varying housing price levels by quantile analysis. Equation (2) is adapted as follows. $Q_\tau$, $\alpha_\tau$ and $\lambda_\tau$, respectively, represent the regression parameter of $\tau$th quantile in the dependent variable, constant term, and control variables. The coefficient $\beta_{0\tau}$ indicates the economic effects of urban regeneration at the $\tau$th quantile of housing price. This paper intends to investigate the effects at the 15th, 20th, ..., 80th, and 85th percentiles of housing prices from low to high.

$$Q_\tau(P_{ijt}) = \alpha_\tau + \beta_{0\tau} * DD_{ijt} + \lambda_\tau Z_{it} + v_i + u_t + \varepsilon_{it} \tag{2}$$

Referring to Zheng, Li, Zheng and Lv [55], $Treat_i^n$, a 0,1-variable, is set to capture the change of effects with the increase in distance. The transaction within 1000 m of urban regeneration projects is divided into 5 groups: 0~200 m, 200~400 m, ..., and 800~1000 m. For example, $Treat_{ijt}^1 = 1$ if the property $i$ is located within 200 m of the project $j$, otherwise, $Treat_{ijt}^1 = 0$. $Post_{ijt}$ takes value 1 if housing $j$ is transacted after the completion of project $j$. The interactive variable of $Treat_{ijt}^n$ and $Post_{ijt}$ replaces the dummy variable $DD_{ijt}$, and captures the effects on housing $i$ within a specific range. The coefficient $\mu_n$ is the estimation of the effects on economic vitality. Thus, the model can be derived as Equation (3).

$$P_{ijt} = \alpha + \mu_n * Treat_{ijt}^n * Post_{ijt} + \lambda Z_{it} + v_i + u_t + \varepsilon_{it} \quad (n = 1, 2, 3, \ldots) \tag{3}$$

There are three regeneration approaches: rehabilitation, revitalization, and redevelopment. The superscript $m$ of the dummy variable $Method_{ij}^m$ takes value 1, 2, and 3 to represent three approaches. If $Method_{ij}^1 = 1$ if the urban regeneration project $j$ is regenerated through rehabilitation, otherwise 0. The definition is also adopted to $Method_{ij}^2$ and $Method_{ij}^3$. The model is expanded as Equation (4). The coefficient $\rho_m$ is the estimation of economic effects by different regeneration approaches.

$$P_{ijt} = \alpha + \beta_0 * DD_{ijt} + \sum_{m=1}^{3} \rho_m * Method_{ij}^m * DD_{ijt} + \sum_{m=1}^{3} \omega_m * Method_{ij}^m + \lambda Z_{it} + v_i + u_t + \varepsilon_{it} \tag{4}$$

The variation of economic effects with the amount of investment is evaluated by the interaction of $DD_{ijt}$ and $Invest_{ij}$. It is a continuous variable added into Equation (1), denoting the investment of the project $j$ which is close to the house $i$. This paper further explores the difference in effects by comparing different regeneration approaches based on Equation (5).

$$P_{ijt} = \alpha + \beta_0 * DD_{ijt} + \varphi * Invest_{ij} * DD_{ijt} + \eta * Invest_{ij} + \lambda Z_{it} + v_i + u_t + \varepsilon_{it} \quad (5)$$

The project location is quantified by the location values. The model is similar to Equation (5), and the coefficient of $DD_{ijt} * Location_{ij}$ indicates the correlation between the project's locations and the effects on surrounding economic vitality. The equation is as follows.

$$P_{ijt} = \alpha + \beta_0 * DD_{ijt} + \zeta * Location_{ij} * DD_{ijt} + \delta * Location_{ij} + \lambda Z_{it} + v_i + u_t + \varepsilon_{it} \quad (6)$$

## 4. Results

### 4.1. Results of Benchmark Regression

Based on Equation (1), this paper assesses the change in housing prices within a 1000 m distance of urban regeneration projects before and after implementation. Table 3 reposts the regression results under different control conditions. In column (1), the coefficient of *DD* without any controls is insignificant, and only 5.6% of samples can be explained. In column (2), controlling for covariables increases the coefficient and model's explanatory power, but the result is still insignificant. In column (3), the fixed effects of time and housing are controlled, and the coefficient is significant −27.8%. The empirical result in column (4) illustrates that urban regeneration has led to an 18.2% reduction in housing prices within the buffer zone significantly when the house-specific characteristic and time effects are controlled, and the value of R-squared reaches 92.3%.

**Table 3.** The overall effects of urban regeneration on surrounding economic vitality.

|  | (1) | (2) | (3) | (4) |
|---|---|---|---|---|
| DD | −0.100 | 0.034 | −0.278 *** | −0.182 *** |
|  | (0.106) | (0.101) | (0.043) | (0.041) |
| POST | 1.484 *** | 1.659 *** | 0.001 | −0.041 |
|  | (0.056) | (0.056) | (0.032) | (0.030) |
| TREATED | −0.927 *** | −0.813 *** | −1.443 *** | −1.461 *** |
|  | (0.075) | (0.077) | (0.096) | (0.075) |
| Area |  | −0.002 |  | −0.004 |
|  |  | (0.001) |  | (0.002) |
| Dec |  | −0.020 |  | 0.687 *** |
|  |  | (0.043) |  | (0.022) |
| Bed |  | 0.482 *** |  | 0.009 |
|  |  | (0.059) |  | (0.076) |
| Ele |  | 2.534 *** |  | 0.337 *** |
|  |  | (0.079) |  | (0.093) |
| Age |  | −0.030 *** |  | −0.045 *** |
|  |  | (0.004) |  | (0.004) |
| Pr |  | 0.017 |  | 0.829 |
|  |  | (0.016) |  | (1.224) |
| Gr |  | 3.773 *** |  | 28.770 *** |
|  |  | (0.351) |  | (9.199) |
| Dissub |  | −0.371 *** |  | −0.454 |
|  |  | (0.059) |  | (0.690) |
| Dise |  | 0.176 *** |  | −0.634 ** |
|  |  | (0.016) |  | (0.299) |
| Disc |  | −0.244 *** |  | 0.311 |
|  |  | (0.041) |  | (0.491) |
| Dispro |  | −0.070 *** |  | −0.080 *** |
|  |  | (0.010) |  | (0.029) |
| House FE | No | No | Yes | Yes |
| Time FE | No | No | Yes | Yes |
| Control Variables | No | Yes | No | Yes |
| R-squared | 0.056 | 0.138 | 0.912 | 0.923 |
| Observations | 21316 | 21316 | 21316 | 21316 |

**Note:** **, and *** mean significant at the level of 5%, and 1%, respectively. House FE and Time FE indicate the housing and time fixed effect. Yes and No indicate the variables are controlled or not.

This paper also conducts several robustness tests to validate the benchmark regression results, including eliminating outliers, changing the range of treatment groups, and so on. In addition to these tests, the Propensity Score Matching (PSM) model is also employed to avoid selection bias on account of observable variables. As a whole, all the tests confirm the robustness of benchmark regression results. Please refer to Appendix C for more information on these analyses.

### 4.2. The Effects on Economic Vitality Considering Different Percentiles of Housing Price Levels

The effects of urban regeneration on surrounding economic vitality, given different levels of housing prices, are explored through quantile regression. The housing price level is categorized by the distribution of housing prices in 15%, 25%, . . ., 75%, and 85% from low to high percentiles. Table 4 depicts the estimated coefficient on different percentiles of housing prices. The effects of urban regeneration are significantly positive only at a 15% level. At the 55% percentile, the economic effects become significantly negative, and such effects increase with the quantile approaching 1. It indicates that the areas with lower housing prices are more sensitive to the benefits of environmental and facility improvement brought by urban regeneration. High-priced housing usually has better infrastructure and physical condition, and urban regeneration can contribute much less to the improvement of the living environment.

**Table 4.** The effects of urban regeneration on economic vitality in view of different percentiles of housing price levels.

|  | (1) | (2) | (3) | (4) | (5) | (6) | (7) | (8) |
|---|---|---|---|---|---|---|---|---|
|  | 15% | 25% | 35% | 45% | 55% | 65% | 75% | 85% |
| DD | 0.234 *** | −0.017 | −0.010 | 0.032 | −0.084 ** | −0.779 *** | −0.933 *** | −1.570 *** |
|  | (3.894) | (−0.087) | (−0.072) | (0.124) | (−2.268) | (−8.218) | (−5.869) | (−67.885) |
| POST | 1.790 *** | 1.401 *** | 1.202 *** | 1.136 *** | 0.766 *** | 1.199 *** | 0.099 | 0.895 *** |
|  | (88.562) | (24.354) | (27.695) | (9.107) | (25.321) | (113.926) | (0.746) | (257.018) |
| TREATED | −0.473 *** | 0.104 ** | −0.070 * | 0.107 | −0.844 *** | −0.851 *** | 2.068 *** | 0.707 *** |
|  | (−20.807) | (2.179) | (−1.920) | (0.388) | (−96.431) | (−44.687) | (7.021) | (37.022) |
| House FE | Yes | Yes | Yes | Yes | Yes | Yes | Yes | Yes |
| Time FE | Yes | Yes | Yes | Yes | Yes | Yes | Yes | Yes |
| Control Variables | Yes | Yes | Yes | Yes | Yes | Yes | Yes | Yes |
| Observations | 21316 | 21316 | 21316 | 21316 | 21316 | 21316 | 21316 | 21316 |

Note: *, **, and *** mean significant at the level of 10%, 5%, and 1%, respectively. House FE and Time FE indicate the housing and time fixed effect. Yes and No indicate the variables are controlled or not.

### 4.3. The Effects on Economic Vitality Considering Distance Gradient

From the center of urban regeneration projects outward, the housing transaction data have been divided into five groups. Table 5 shows the regression results of economic effects regarding each distance gradient. The significant negative effects are observed only between 400 m to 800 m. This paper also calculates the influence on the 1000 m to 2000 m range. The curve of economic effects is plotted in Figure 2. It shows that the negative effects become insignificant when the distance exceeds 800 m. This paper further changes the scope of control groups to 1–2 km, 2–3 km, and 3–4 km. The range of treatment groups remains the same. Based on Equation (3), the regression results are reported in Table 6. In column (2), the coefficient is less than the benchmark results, but it is much greater when the range becomes 3–4 km in columns (4) and (6). It indicates that the effects of urban regeneration on surrounding economic vitality decrease with the increase in distance to projects.

**Table 5.** The effects of urban regeneration on economic vitality within different distance gradients.

| | (1) | (2) | (3) | (4) | (5) |
|---|---|---|---|---|---|
| | 0–200 m | 200–400 m | 400–600 m | 600–800 m | 800–1000 m |
| $DD_{02}$ | −0.038 (0.092) | | | | |
| $DD_{04}$ | | −0.137 (0.085) | | | |
| $DD_{06}$ | | | −0.403 *** (0.066) | | |
| $DD_{08}$ | | | | −0.177 ** (0.073) | |
| $DD_{10}$ | | | | | 0.127 (0.084) |
| POST | −0.083 *** (0.028) | −0.079 *** (0.028) | −0.060 ** (0.029) | −0.077 *** (0.029) | −0.092 *** (0.028) |
| House FE | Yes | Yes | Yes | Yes | Yes |
| Time FE | Yes | Yes | Yes | Yes | Yes |
| Control Variables | Yes | Yes | Yes | Yes | Yes |
| R-squared | 0.923 | 0.923 | 0.923 | 0.923 | 0.923 |
| Observations | 21316 | 21316 | 21316 | 21316 | 21316 |

**Note:** **, and *** mean significant at the level of 5%, and 1%, respectively. House FE and Time FE indicate the housing and time fixed effect. Yes and No indicate the variables are controlled or not.

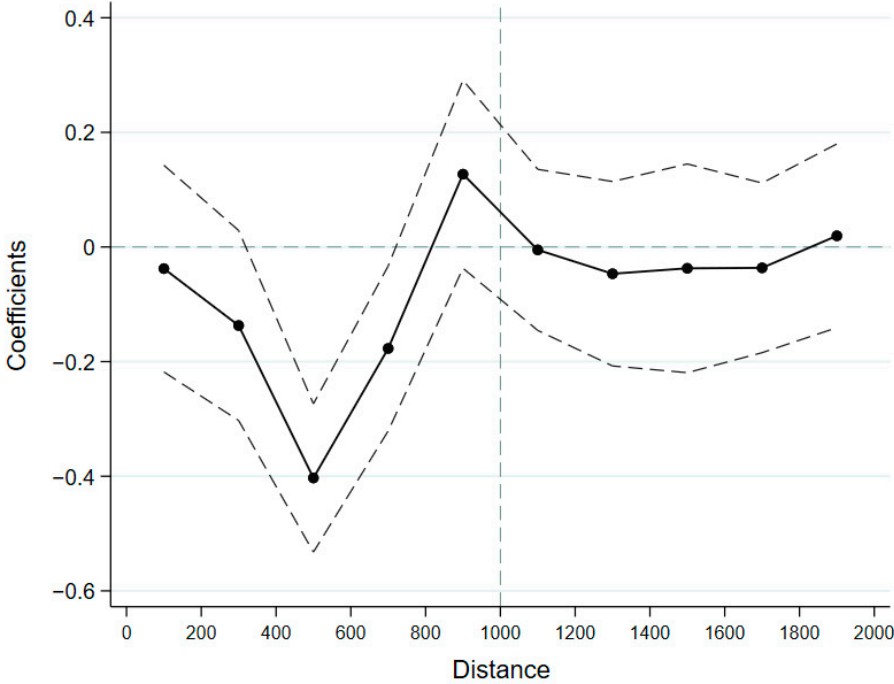

**Figure 2.** The effects of urban regeneration on economic vitality considering different distance gradients. The solid line indicates the changing tendency of economic effects within different distance gradient. The dotted line is the 95% confidence interval.

**Table 6.** The effects of urban regeneration on economic vitality considering different distance gradients in view of the control groups.

| | (1) | (2) | (3) | (4) | (5) | (6) |
|---|---|---|---|---|---|---|
| | 0~2 km | | 0~3 km | | 0~4 km | |
| DD | −0.174 *** | −0.123 ** | −0.236 *** | −0.204 *** | −0.502 *** | −0.413 *** |
| | (0.052) | (0.049) | (0.062) | (0.056) | (0.085) | (0.076) |
| POST | −0.012 | −0.061 | 0.082 | 0.042 | 0.308 *** | 0.233 *** |
| | (0.049) | (0.046) | (0.060) | (0.055) | (0.083) | (0.074) |
| TREATED | −0.725 | −9.784 | 1.644 *** | −57.780 *** | 1.709 *** | −7.796 *** |
| | (0.692) | (19.510) | (0.268) | (5.594) | (0.449) | (2.982) |
| Treatment Groups | 0~1 km | | 0~1 km | | 0~1 km | |
| Control Groups | 1~2 km | | 2~3 km | | 3~4 km | |
| House FE | Yes | Yes | Yes | Yes | Yes | Yes |
| Time FE | Yes | Yes | Yes | Yes | Yes | Yes |
| Control Variables | No | Yes | No | Yes | No | Yes |
| R-squared | 0.919 | 0.928 | 0.919 | 0.930 | 0.917 | 0.929 |
| Observations | 11816 | 11816 | 9008 | 9008 | 7381 | 7381 |

**Note:** **, and *** mean significant at the level of 5%, and 1%, respectively. House FE and Time FE indicate the housing and time fixed effect. Yes and No indicate the variables are controlled or not.

### 4.4. The Effects on Economic Vitality Considering Regeneration Approach

The effects on economic vitality considering different regeneration approaches are shown in Panel A of Table 7. The treatment effects of the three approaches are estimated based on Equation (4). The interaction variable's coefficient, $DD_{ijt} * Method^m$, is the approximate estimation of effects. The results show that all types of urban regeneration projects have negative effects. The housing prices close to revitalization projects decreased by 29.3%, much more than other projects. The decreases for rehabilitation and redevelopment projects are 19.1% and 18.4%, respectively.

**Table 7.** The effects of urban regeneration on economic vitality with different generation approaches.

| | (1) | (2) | (3) |
|---|---|---|---|
| | Rehabilitation | Revitalization | Redevelopment |
| | Panel A: Average effects | | |
| DD×Method[1] | −0.191 *** | | |
| | (0.044) | | |
| DD×Method[2] | | −0.293 ** | |
| | | (0.120) | |
| DD×Method[3] | | | −0.184 ** |
| | | | (0.076) |
| House FE | Yes | Yes | Yes |
| Time FE | Yes | Yes | Yes |
| Control Variables | Yes | Yes | Yes |
| R-squared | 0.923 | 0.923 | 0.923 |
| Observations | 21316 | 21316 | 21316 |
| | Panel B: Effects within different distance gradient | | |
| 0–200 m | −0.036 | −0.171 | −0.125 |
| | (0.109) | (0.307) | (0.176) |
| 200–400 m | −0.081 | −0.489 ** | −0.336 * |
| | (0.099) | (0.243) | (0.184) |

**Table 7.** *Cont.*

|  | (1) | (2) | (3) |
|---|---|---|---|
|  | **Rehabilitation** | **Revitalization** | **Redevelopment** |
| 400–600 m | −0.345 *** | −0.474 ** | −0.548 *** |
|  | (0.080) | (0.206) | (0.122) |
| 600–800 m | −0.235 *** | −0.323 | 0.105 |
|  | (0.083) | (0.228) | (0.198) |
| 800–1000 m | 0.022 | 0.152 | 0.269 * |
|  | (0.104) | (0.276) | (0.152) |
|  |  |  |  |
| House FE | Yes | Yes | Yes |
| Time FE | Yes | Yes | Yes |
| Control Variables | Yes | Yes | Yes |
| R-squared | 0.923 | 0.923 | 0.923 |
| Observations | 21316 | 21316 | 21316 |

**Note:** *, **, and *** mean significant at the level of 10%, 5%, and 1%, respectively. House FE and Time FE indicate the housing and time fixed effect. Yes and No indicate the variables are controlled or not.

The spatial heterogeneity of different regeneration approaches is presented in Panel B of Table 7. For better comparison, the change curves are plotted in Figure 3. It indicates that (1) the effects of all three types of projects are significantly negative. (2) the effects of rehabilitation-type projects rank in the middle and are significant within the range of 400–800 m. (3) The revitalization-type projects hold the most potent negative effects, with a significant influence range between 200–600 m. (3) The redevelopment-type projects have minimum effects. Their negative influence is significant within 200–600 m, but the positive influence can be observed in 800–1000 m.

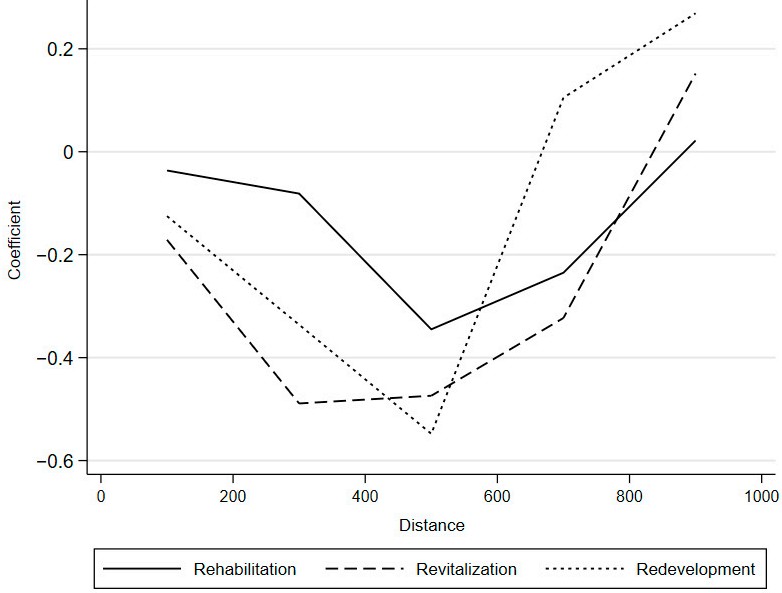

**Figure 3.** The effects of three regeneration approaches on economic vitality considering different distance gradients.

### 4.5. The Effects on Economic Vitality Considering Investment Scale

The effects of urban regeneration on surrounding economic vitality considering the project investment are investigated based on Equation (5). Table 8 reports the regression results. In column (1), all the housing transactions samples are used. The coefficient of the interaction variable is significant, while the estimation of *DD* is close to the benchmark result. It indicates that an increase in investment leads to a decrease in the effects. Columns (2), (3), and (4) show that only the revitalization-type projects significantly contribute to the decline of nearby housing prices with the growth of project investment.

The increasing investment in rehabilitation and redevelopment-type projects leads to an insignificant change in nearby housing prices. To some extent, the results can be regarded as the return on investment of urban regeneration.

**Table 8.** The effects of urban regeneration on economic vitality considering investment.

|  | (1) | (2) | (3) | (4) |
|---|---|---|---|---|
|  | **All** | **Rehabilitation** | **Revitalization** | **Redevelopment** |
| DD×Investment | −0.024 * | 0.129 | −0.043 *** | 0.006 |
|  | (0.012) | (0.093) | (0.013) | (0.023) |
| DD | −0.174 *** | −0.224 *** | −0.180 *** | −0.202 *** |
|  | (0.040) | (0.041) | (0.039) | (0.039) |
| Investment | −0.413 *** | −0.252 | −0.330 ** | −0.419 *** |
|  | (0.045) | (0.219) | (0.138) | (0.053) |
|  |  |  |  |  |
| House FE | Yes | Yes | Yes | Yes |
| Time FE | Yes | Yes | Yes | Yes |
| Control Variables | Yes | Yes | Yes | Yes |
| R-squared | 0.923 | 0.923 | 0.923 | 0.923 |
| Observations | 21316 | 21316 | 21316 | 21316 |

**Note:** *, **, and *** mean significant at the level of 10%, 5%, and 1%, respectively. House FE and Time FE indicate the housing and time fixed effect. Yes and No indicate the variables are controlled or not.

### 4.6. The Effects on Economic Vitality Considering Location Values

The effects of urban regeneration on surrounding economic vitality, considering the project location values, are probed based on Equation (6). The results are listed in Table 9. The overall effects, considering location values, are significantly negative. Such significant negative effects hold across the 25th, 50th, and 75th percentile of location values. It implies that urban regeneration projects in bad locations lead to a slighter decrease in nearby housing prices. Given different regeneration approaches, the revitalization and redevelopment-type projects have significant negative effects considering location values. The former is stronger than the latter. Meanwhile, the changes of negative effects of all types of projects with increased location value quantile are similar to the overall results of full samples.

**Table 9.** The effects of urban regeneration on economic vitality considering location values.

|  | (1) | (2) | (3) | (4) |
|---|---|---|---|---|
|  | **All** | **Rehabilitation** | **Revitalization** | **Redevelopment** |
| DD×Location | −0.191 * | −0.174 | −1.131 *** | −0.596 *** |
|  | (0.3.19) | (0.299) | (0.403) | (0.390) |
| DD | −0.176 *** | −0.177 *** | −0.177 *** | −0.178 *** |
|  | (0.069) | (0.049) | (0.039) | (0.041) |
| Location | 6.024 | 1.602 | 5.022 | 0.235 |
|  | (17.700) | (15.650) | (17.440) | (15.300) |
|  |  |  |  |  |
| $\beta_0 * DD_{jt} + \zeta * Location_j * DD_{jt}$ |  |  |  |  |
| At the 25th percentile | −0.190 *** | −0.192 *** | −0.264 *** | −0.224 *** |
|  | (0.051) | (0.039) | (0.044) | (0.041) |
| At the 50th percentile | −0.206 *** | −0.209 *** | −0.360 *** | −0.275 *** |
|  | (0.041) | (0.042) | (0.067) | (0.062) |
| At the 75th percentile | −0.229 *** | −0.232 *** | −0.495 *** | −0.346 *** |
|  | (0.055) | (0.065) | (0.110) | (0.103) |
|  |  |  |  |  |
| House FE | Yes | Yes | Yes | Yes |
| Time FE | Yes | Yes | Yes | Yes |
| Control Variables | Yes | Yes | Yes | Yes |
| R-squared | 0.923 | 0.923 | 0.923 | 0.923 |
| Observations | 21316 | 21316 | 21316 | 21316 |

**Note:** *, and *** mean significant at the level of 10%, and 1%, respectively. House FE and Time FE indicate the housing and time fixed effect. Yes and No indicate the variables are controlled or not.

## 5. Discussion

### 5.1. The Reflections on the Effects of Urban Regeneration on Surrounding Economic Vitality

The study demonstrates that the DID method is an effective way to measure the treatment effects, thus providing an insight into the effects of urban regeneration on surrounding economic vitality. Since the sensitivity and representation of housing prices to economic vitality, the research examines the change in surrounding housing prices to capture the effects of urban regeneration on economic vitality. In addition, a series of measures is applied to improve the robustness of benchmark regression results. It considers the market restriction during the COVID-19 pandemic, the impact of outlier data on housing prices, the long period of project implementation, the distance radius in treatment and control groups, and the possible endogeneity in the designation of urban regeneration areas. All the tests are passed, and the benchmark regression results are confirmed.

Overall, it is surprising to find the negative effects of urban regeneration in the Chinese context. It contradicts many research studies that believe urban regeneration can bring economic benefits by increasing property values in the surrounding areas [34,73,74]. Nevertheless, the negligible effects on economic outcomes and growth are also supported by the research on industrial site regeneration in the Netherlands [21] and urban regeneration policies in Italy [34]. Meanwhile, a few works focusing on specific types of projects in different backgrounds conclude negative effects [50,54]. According to the studies in new construction field, this may be due to the specific market scale in China [75]. The real estate market has experienced a skyrocketing period for a long time. Today, the market is staying on high ground, and the housing supply and demand have reached a relative balance. Urban regeneration can bring about a new supply of high-quality urban housing, yet the housing demand in a certain urban area may remain the same for a short time. In this situation, in the region, the potential housing transactions are more likely to occur in the regenerated neighborhood rather than its surrounding properties (especially second-hand houses). Therefore, the housing prices in the area surrounding urban regeneration projects decrease after project implementation.

The results also show that the economic effect of urban regeneration non-monotonically changes with the increased distance to the project. Several existing research studies have demonstrated that the farther the distance, the smaller the effects will be [6,12,47,57]. Nevertheless, the negative effects in this research are significant within 400–800 m only. This monotonical and localized changing trend is because the new facilities and amenities provided by urban regeneration can offer living convenience in a limited range outside the project scope.

This paper demonstrates that higher investment in urban regeneration projects leads to more substantial negative effects. Previous studies on the effects of the renewal policy implemented in Berlin revealed that the public fund can be positively converted into property value in the regenerated projects by indirectly attracting property buyers [6]. Yet, attracting more buyers to the regenerated project means alienating them from the surrounding areas. In China, most urban regeneration projects are mostly government-initiated and funded. This research further reveals that higher-level government investment in an urban regeneration project may reduce market attraction in its nearby properties.

It has also been proved that better project location leads to stronger negative effects. The results can be explained as the law of diminishing marginal utility [76]. Bad project location means worse existing living quality and the need for more public facilities. Thus, the area with a lower location value (lower housing price) can benefit more from improving the surrounding living environment. In comparison, urban regeneration can contribute much less to improving the area with better location values. However, previous studies revealed the poor capability of risk response and economic recovery to real estate market crisis, bringing about less private investment to improve living conditions in regions with cheap properties [77]. Comparing the ignorance of such projects in property-led redevelopment, sustainable urban regeneration places a greater emphasis on residential

needs and public requirements [78]. That is to say, under limited resources, it becomes more necessary and effective to prioritize urban regeneration projects in bad locations not only for the need for economic revitalization but also for social equity.

### 5.2. The Comparison between Different Regeneration Approaches

Although some research studies have explored the economic changes brought by specific types of urban regeneration projects in a different context, they lack systematic comparison between the different projects with diverse attributes, not to mention in the Chinese context. In addition to the overall effects assessment, this paper also contributes to filling the above research gap. Compared to redevelopment and rehabilitation-type projects, revitalization has the most significant negative effects. In China, much revitalization occurred in the areas with historical value. The results indicate that the revitalization may bring about more significant improvement in the targeted neighborhoods. Thus, the buyers in the area are more interested in the revitalized neighborhoods rather than their surrounding properties. Some scholars believe that the housing market will be affected by the revitalization-type projects in historical sites due to the strict restriction of urban development in the surrounding area [41]. Such restrictions include building density, building height, and development intensity. The research on heritage conservation in Auckland concludes similarly: comparing the price premium of houses with less restriction, property value with strong building regulations decreased significantly [12]. Furthermore, other research studies have pointed out that investing in small-scale revitalization-type projects is not efficient from an area's point of view [12,79]. However, this research found that the effects of revitalization-type projects decrease with the growth of the investment scale. That is to say, from the perspective of overall area development, investing more in revitalization is unnecessary since it cannot generate better effects. In addition, considering project location, conducting revitalization in an area with low location value is much more meaningful compared to other types of projects.

In comparison, the negative effects of redevelopment-type projects on surrounding economic vitality are minimal, and only significantly affects the property within a 400–800 m radius. According to other research studies, redevelopment generally exerts a positive influence on surrounding property prices in the long term [13,47]. In this research, the negative effects after project implementation may be due to the long implementation period. Compared to other projects, redevelopment involves the demolition and reconstruction phases, taking much longer in the whole procedure. The negative effects are generated by the severe environmental impact. On the one hand, the pollution produced by demolition and construction work affects the surrounding living environment. On the other hand, redevelopment always refers to the improvement of the plot ratio. The growing building height and density may have an impact on the direct sunlight and view of nearby houses [54]. Furthermore, large investments generally represent large redevelopment scales and huge improvements in the area's environment [13,73]. However, it is surprising to find that the investment scale of redevelopment has no significant influence on the change of effects. Moreover, the higher location values relate to stronger negative effects. This contradicts some existing research studies about redevelopment in other backgrounds [39]. It further illustrates that different cultural, institutional, and market environments lead to diverse effects of urban regeneration on the economic vitality of the surrounding area.

The effects of rehabilitation-type projects are more petite than revitalization-type projects but more remarkable than redevelopment-type projects. Generally, rehabilitation occurs in dilapidated old neighborhoods without historical value. It improves the living quality of the targeted neighborhood with small-scale physical change. A few studies focus on the economic effects of urban rehabilitation in the American context, generally drawing positive conclusions [20,80,81]. Nonetheless, comparing most private renovation internationally, in China, the investment strongly relies on government funding, since this type of urban regeneration often fails to generate sufficient market value on its own. Therefore, once a neighborhood in an area is designated as a rehabilitation-type project,

it is almost impossible for the other neighborhoods in this area to be rehabilitated in the near future. Previous research studies have demonstrated that the rehabilitation of one old neighborhood can improve the willingness of surrounding residents to rehabilitate their neighborhoods [81]. However, in the Chinese context, few surrounding residents are eager to pay by their own due to the sense of inequality. This is why rehabilitation leads to negative effects. In addition, although the positive growth of effects caused by the increase in investment in rehabilitation is much higher than that of redevelopment-type projects, it is still insignificant. As found by Rossi-Hansberg, Sarte and Owens [7], the land value of a rehabilitated neighborhood can increase by $2 to $6 per dollar invested. This is the possible explanation for the positive correlation between investment and the effects of rehabilitation-type projects.

## 6. Conclusions

Urban regeneration is a sustainable approach to dealing with urban decay and improving the living environment. In addition to bringing about direct benefits in the regenerated area, different types of urban regeneration projects also exert indirect economic influence on the surrounding area, namely, external effects. This paper aims to evaluate the effects of urban regeneration and compare the differences between diverse types of projects in the Chinese context based on the change in housing prices. Chongqing is selected as the case city, and 37 urban regeneration projects with diverse attributes are chosen as experiment subjects. The staggered DID method is employed to capture the change in housing prices nearby urban regeneration projects. Overall, urban regeneration significantly affects surrounding economic vitality, which is heterogeneous with different project types, especially under the regeneration approach. The main findings of this paper are as follows: (1) Urban regeneration leads to an 18.2% decline in housing prices within the range of 1000 m surrounding urban regeneration projects. The negative effects are most remarkable within the 400–600 m range. (2) Revitalization-type projects have the greatest negative effects, while rehabilitation-type and redevelopment-type projects have similar effects. (3) The increase in investment scale leads to a negative change in the effects of urban regeneration, especially for revitalization-type projects. (4) A better project location leads to more substantial negative effects, but the negative correlation between location and urban regeneration is insignificant for rehabilitation-type projects.

This paper contributes to providing insights into the urban regeneration assessment from the perspective of economic effects. It involves different regeneration approaches in comparison to meet the practical demand, and provides valuable references for the government policymakers and urban planners to make better and more sustainable decisions. Moreover, the research paradigm of "out-looking" project evaluation can be adopted in another field regarding urban development. This paper demonstrates that the effects of urban regeneration appear differently given investment scale and project location. According to the findings, the policy implications are concluded and proposed as follows. (1) More priority can be given to rehabilitation or redevelopment-type projects. (2) Investing in projects with worse location values can bring more positive overall benefits. Yet, increasing the investment scale of revitalization-type projects is of less significance.

Finally, there is still some room for improvement in this research. Firstly, although Chongqing is a pioneer city in promoting urban regeneration, only studying the Chongqing case cannot determine every scenario in China. Future research will try to involve more representative cities to obtain a holistic view. Secondly, housing prices are one of the most representative indicators of economic effects, but it does not mean other indicators are meaningless. In the future, more indicators, such as nearby commercial types and consumption levels, will be evaluated. Finally, due to the lack of available data, a detailed examination of the time-series effects that span multiple stages of urban regeneration implementation could not be conducted in detail in this study. With the increasing emphasis on life circle analysis in sustainable urban regeneration [82,83], the dynamic effects evaluation on different regeneration phases is recommended.

**Author Contributions:** Conceptualization, M.Y. and H.W.; methodology, H.W.; software, M.Y.; formal analysis, M.Y. and H.W.; resources, H.W.; writing—original draft preparation, M.Y.; writing—review and editing, H.W.; visualization, M.Y.; supervision, H.W.; funding acquisition, H.W. All authors have read and agreed to the published version of the manuscript.

**Funding:** This research was funded by the National Natural Science Foundation of China (Grant No.72301044); Chongqing Social Sciences Planning Project—Youth Fund (Grant No.2022NDQN43).

**Data Availability Statement:** The data presented in this study are available on request from the corresponding author. The data are not publicly available to protect the privacy of the study's participants.

**Conflicts of Interest:** The authors declare no conflicts of interest.

## Appendix A. Calculation of Urban Regeneration Projects' Location Value

**Table A1.** The weight of distance to different public facilities based on entropy weight method.

| Type | Facility | Entropy Weight |
|---|---|---|
| Transportation | Subway station | 10.40% |
| | Bus station | 17.71% |
| Administration | Government agency | 15.35% |
| Culture and Sports | Gym, museum, library, gallery, and so on | 12.95% |
| Commerce | Shopping mall | 11.84% |
| Finance and Post | Bank and post office | 11.55% |
| Education | College | 7.82% |
| Health care | General Hospital | 12.37% |

**Table A2.** The details of locational value calculation of urban regeneration projects based on TOPSIS.

| Projects | $D_i^+$ | $D_i^-$ | $C_i$ | Rank |
|---|---|---|---|---|
| LJC | 578.067 | 52.614 | 0.083 | 28 |
| TSGC | 535.868 | 102.426 | 0.160 | 23 |
| NRC | 376.980 | 428.854 | 0.532 | 2 |
| CSC | 542.945 | 105.148 | 0.162 | 22 |
| QC | 374.235 | 314.128 | 0.456 | 8 |
| TFYC | 590.108 | 48.970 | 0.077 | 32 |
| TFJC | 559.131 | 77.582 | 0.122 | 25 |
| TTJC | 568.947 | 71.779 | 0.112 | 26 |
| RC | 443.622 | 244.253 | 0.355 | 11 |
| FRC | 446.902 | 212.961 | 0.323 | 12 |
| LC | 455.027 | 202.828 | 0.308 | 13 |
| SC | 587.838 | 46.844 | 0.074 | 34 |
| JC | 580.933 | 52.635 | 0.083 | 30 |
| KNC | 566.508 | 60.518 | 0.097 | 27 |
| KC | 523.984 | 108.351 | 0.171 | 21 |
| ZLC | 536.616 | 118.147 | 0.180 | 20 |
| SRC | 515.576 | 145.624 | 0.220 | 18 |
| DC | 602.737 | 45.820 | 0.071 | 35 |
| KLC | 590.200 | 40.955 | 0.065 | 36 |
| XQRC | 474.501 | 185.038 | 0.281 | 15 |
| WC | 403.300 | 353.288 | 0.467 | 6 |
| YC | 390.742 | 343.010 | 0.467 | 5 |
| CVS | 499.283 | 147.892 | 0.229 | 17 |
| FDSS | 601.515 | 49.976 | 0.077 | 31 |
| SHSS | 234.076 | 533.323 | 0.695 | 1 |
| YSS | 350.695 | 359.533 | 0.506 | 3 |
| YJSS | 581.257 | 52.771 | 0.083 | 29 |
| SRSS | 413.009 | 289.126 | 0.412 | 9 |
| SFSS | 585.109 | 47.098 | 0.074 | 33 |

**Table A2.** *Cont.*

| Projects | $D_i^+$ | $D_i^-$ | $C_i$ | Rank |
|----------|---------|---------|-------|------|
| LSS | 407.090 | 355.406 | 0.466 | 7 |
| XMSS | 588.248 | 102.701 | 0.149 | 24 |
| SHD | 488.914 | 192.699 | 0.283 | 14 |
| MCLHD | 487.284 | 173.486 | 0.263 | 16 |
| IMP | 607.649 | 30.843 | 0.048 | 37 |
| RTC | 364.861 | 331.060 | 0.476 | 4 |
| ICCIP | 511.663 | 118.770 | 0.188 | 19 |
| IM | 403.938 | 251.933 | 0.384 | 10 |

**Note:** $D_i^+$ indicates the distance from urban regeneration project $i$ to the positive ideal solution, while $D_i^-$ indicates the distance from urban regeneration project $i$ to the negative ideal solution. $C_i$ indicates the relative degree of approximation. Its value is between 0 and 1, and the projects $i$ with highest value is ranked as No. 1.

## Appendix B. Premised Test

In this research, the empirical analysis is based on the assumption that urban regeneration decision and planning is not affected by nearby housing prices. Thus, according to previous literature, the survival analysis is utilized to check their correlation [84]. It tests whether the average housing price within 1000 m of regenerated areas determines it is designated to be regenerated. The control variables include project-specific characteristics and social-economic characteristics. The assumption holds when the results are insignificant. As shown in Table A3, the coefficients of Price under all conditions are insignificant when controlling any variables. It demonstrates that the change in housing price within 1000 m of these regenerated areas cannot influence whether they are designated to be regenerated. The results support the assumption that the start time of urban regeneration projects is not affected by nearby housing prices.

**Table A3.** The correlation between average housing price and the start time of urban regeneration.

| | (1) | (2) | (33) | (4) |
|---|---|---|---|---|
| Price | 0.066 | 0.064 | 0.111 | −0.168 |
| | (0.081) | (0.270) | (0.110) | (0.123) |
| Floor area | | −0.006 | | 0.005 |
| | | (0.010) | | (0.009) |
| Accessibility of Public Facilities | | 209.7 | | 350.600 *** |
| | | (265.600) | | (86.85) |
| Equity of Public Facilities | | −8.377 | | −87.760 *** |
| | | (8.227) | | (15.71) |
| Population Density | | | −27.460 | 311.200 *** |
| | | | (27.740) | (60.700) |
| Development Intensity | | | 0.181 | −0.756 ** |
| | | | (0.311) | (0.301) |
| Observations | 149 | 114 | 149 | 114 |

**Note:** **, and *** mean significant at the level of 5%, and 1%, respectively. Yes and No indicate the variables are controlled or not.

The parallel trend assumption (PTA) is the premise for estimating treatment effects by DID method. The parallel trend test is similar to the event study, and the equation is as follows.

$$\text{P}_{ijt} = \alpha + \beta_1 \text{DD}_{ijt}^{-13} + \beta_2 \text{DD}_{ijt}^{-12} + \ldots + \beta_{23} \text{DD}_{ijt}^{+10} + \beta_{24} \text{DD}_{ijt}^{+11} + \lambda Z_{it} + v_i + u_t + \varepsilon_{it} \quad \text{(A1)}$$

where the dummy variable $DD_{ijt}^{-m}$ or $DD_{ijt}^{+m}$ equals 1 when the property $i$ within 1000 m of projects is transacted in the $m$th half year before or after urban regeneration, otherwise is 0. The project implementation period is excluded to avoid multicollinearity. The coefficient $\beta_m$ is the difference between control groups and treatment groups.

Figure A1 plots the results of five periods before and after urban regeneration with 95% confidence intervals. The coefficients of pretreatment variables are positive insignificantly. It illustrates that in statistics, the variation in housing price is approximatively regarded as the result of urban regeneration rather than other reasons. After urban regeneration, the housing price shows a significant downward trend.

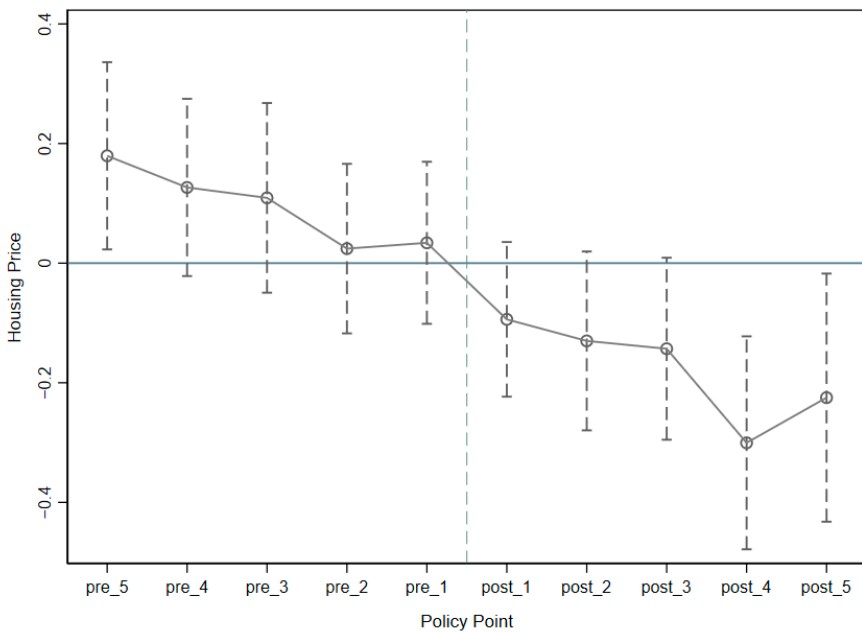

**Figure A1.** The changing tendency of housing price.

**Appendix C. Robustness Test**

This paper also conducts several robustness tests to confirm the benchmark regression results. Firstly, it has been demonstrated that at the early stage of the COVID-19 pandemic, significant fluctuations existed in the real estate market in China [62,85]. The housing transaction is only allowed online until Marth 9th, when the pandemic is under control. To avoid the biased estimation of economic impacts, Equation (1) is re-run after excluding the transaction data in the first half year of 2020. Secondly, the housing transaction near four projects (i.e., NRC, SFSS, LSS, and SHD) are excluded since their implementation periods are longer than 30 months. The remaining data is still regressed based on Equation (1). Thirdly, since the outlier data of housing price probably affect the results, the model of Equation (1) is re-run after excluding the average housing prices below the 1st and above the 99th percentiles. Finally, the range of treatment groups is changed from 1000 m to 2000 m, and the control groups are divided into three buffer zones within the different radii.

The results of four robustness tests are shown in the appendix. Table A4 reports the results by, respectively, excluding the transactions in the first half of 2020 and transactions close to the projects with a long implementation period. The coefficients almost have no disparity with the above results. The coefficients in Table A5 are the results by excluding outliers below the 1st and above the 99th percentiles, and they are close to the result of the full sample. After changing the range of treatment groups and control groups, the estimation of treatment effects is slightly different from the baseline results (See Table A6).

**Table A4.** The effects on economic vitality after excluding the influence of COVID-19 and long implementation period.

| | (1) | (2) | (3) | (4) |
|---|---|---|---|---|
| | **Excluding the Influence of COVID-19** | | **Excluding the Influence of 4 Projects** | |
| DD | −0.325 *** (0.046) | −0.220 *** (0.042) | −0.265 *** (0.044) | −0.167 *** (0.042) |
| POST | 0.003 (0.035) | −0.031 (0.032) | −0.006 (0.033) | −0.042 (0.031) |
| TREATED | −1.498 *** (0.062) | −1.464 *** (0.237) | −0.679 (0.694) | −1.585 *** (0.207) |
| Area | | −0.017 *** (0.002) | | −0.004 (0.002) |
| Dec | | 0.658 *** (0.022) | | 0.695 *** (0.022) |
| Bed | | 0.408 *** (0.046) | | 0.000 (0.074) |
| Ele | | 0.308 *** (0.096) | | 0.265 *** (0.097) |
| Age | | −0.047 *** (0.004) | | −0.045 *** (0.004) |
| Pr | | 4.415 *** (1.288) | | 3.082 ** (1.361) |
| Gr | | −44.230 (37.140) | | −5.710 (33.250) |
| Dissub | | −3.161 (3.193) | | −1.264 *** (0.201) |
| Dise | | 0.092 (1.265) | | −0.539 * (0.317) |
| Disc | | −0.257 (1.339) | | −0.217 (0.915) |
| Dispro | | −0.148 (0.261) | | −0.150 (0.236) |
| House FE | Yes | Yes | Yes | Yes |
| Time FE | Yes | Yes | Yes | Yes |
| Control Variables | No | Yes | No | Yes |
| R-squared | 0.915 | 0.927 | 0.911 | 0.921 |
| Observations | 19095 | 19095 | 20026 | 20026 |

**Note:** *, **, and *** mean significant at the level of 10%, 5%, and 1%, respectively. House FE and Time FE indicate the housing and time fixed effect. Yes and No indicate the variables are controlled or not.

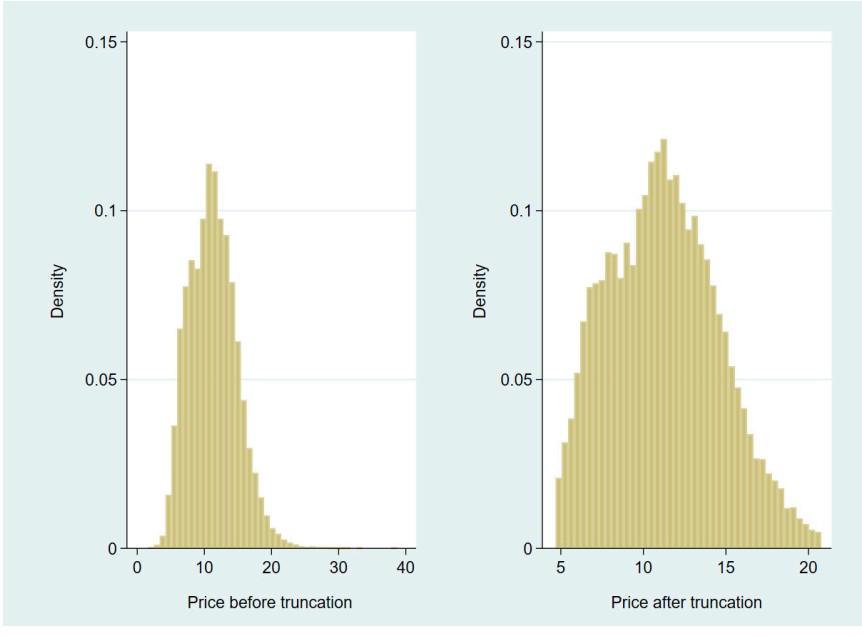

**Figure A2.** The distribution of housing prices before and after data truncation.

**Table A5.** The effects on economic vitality after excluding outliers.

| | (1) | (2) | (3) | (4) |
|---|---|---|---|---|
| | **With Outliers** | **Excluding Percentiles** | | |
| | | **1st** | **99th** | **1st and 99th** |
| DD | −0.182 *** | −0.167 *** | −0.145 *** | −0.130 *** |
| | (0.041) | (0.041) | (0.039) | (0.039) |
| POST | −0.041 | −0.036 | −0.060 ** | −0.055 * |
| | (0.030) | (0.030) | (0.028) | (0.028) |
| TREATED | −1.461 *** | −1.600 *** | −1.626 *** | −1.666 *** |
| | (0.075) | (0.207) | (0.201) | (0.202) |
| Area | −0.004 | −0.004 | −0.003 * | −0.003 * |
| | (0.002) | (0.002) | (0.002) | (0.002) |
| Dec | 0.687 *** | 0.692 *** | 0.657 *** | 0.662 *** |
| | (0.022) | (0.022) | (0.019) | (0.020) |
| Bed | 0.009 | 0.013 | −0.007 | −0.003 |
| | (0.076) | (0.075) | (0.064) | (0.064) |
| Ele | 0.337 *** | 0.345 *** | 0.343 *** | 0.350 *** |
| | (0.093) | (0.095) | (0.089) | (0.091) |
| Age | −0.045 *** | −0.043 *** | −0.037 *** | −0.036 *** |
| | (0.004) | (0.004) | (0.003) | (0.003) |
| Pr | 0.829 | 3.025 ** | 2.811 ** | 2.796 ** |
| | (1.224) | (1.368) | (1.333) | (1.343) |
| Gr | 28.770 *** | −5.678 | −3.138 | −3.035 |
| | (9.199) | (33.300) | (32.500) | (32.650) |
| Dissub | −0.454 | −1.302 *** | −1.338 *** | −1.367 *** |
| | (0.690) | (0.206) | (0.199) | (0.204) |
| Dise | −0.634 ** | −0.512 | −0.477 | −0.463 |
| | (0.299) | (0.319) | (0.311) | (0.313) |
| Disc | 0.311 | −0.195 | −0.098 | −0.089 |
| | (0.491) | (0.918) | (0.896) | (0.902) |
| Dispro | −0.080 *** | −0.147 | −0.142 | −0.140 |
| | (0.029) | (0.236) | (0.231) | (0.232) |
| | | | | |
| House FE | Yes | Yes | Yes | Yes |
| Time FE | Yes | Yes | Yes | Yes |
| Control Variables | Yes | Yes | Yes | Yes |
| R-squared | 0.924 | 0.921 | 0.927 | 0.926 |
| Observations | 21316 | 21104 | 21103 | 20891 |

**Note:** *, **, and *** mean significant at the level of 10%, 5%, and 1%, respectively. House FE and Time FE indicate the housing and time fixed effect. Yes and No indicate the variables are controlled or not.

**Table A6.** The effects on economic vitality of different treatment groups and control groups.

| | **All** | | **0–3 km** | | **0–4 km** | |
|---|---|---|---|---|---|---|
| | **(1)** | **(2)** | **(3)** | **(4)** | **(5)** | **(6)** |
| DD | −0.253 *** | −0.158 *** | −0.146 *** | −0.115 ** | −0.239 *** | −0.176 *** |
| | (0.041) | (0.038) | (0.056) | (0.053) | (0.050) | (0.046) |
| POST | 0.0723 * | 0.002 | 0.024 | −0.027 | 0.075 | 0.011 |
| | (0.037) | (0.034) | (0.056) | (0.052) | (0.048) | (0.044) |
| TREATED | −4.602 *** | −20.820 ** | −0.181 | −24.030 | −0.296 | −12.190 |
| | (0.592) | (10.500) | (0.506) | (21.620) | (0.619) | (15.390) |
| | | | | | | |
| Treatment Groups | 0–2 km | | 0–2 km | | 0–2 km | |
| Control Groups | >2 km | | 2–3 km | | 2–4 km | |
| House FE | Yes | Yes | Yes | Yes | Yes | Yes |
| Time FE | Yes | Yes | Yes | Yes | Yes | Yes |
| Control Variables | No | Yes | No | Yes | No | Yes |
| R-squared | 0.912 | 0.923 | 0.926 | 0.915 | 0.924 | 0.912 |
| Observations | 21316 | 21316 | 15510 | 15510 | 17577 | 17577 |

**Note:** *, **, and *** mean significant at the level of 10%, 5%, and 1%, respectively. House FE and Time FE indicate the housing and time fixed effect. Yes and No indicate the variables are controlled or not.

To some extent, the selection of urban regeneration projects is not a perfect quasi-experiment. The Propensity Score Matching (PSM) is a dominant way to avoid selection bias on account of observable variables and capture more reliable average treatment effects [86]. Its basic idea is to match individuals in treatment groups to those in nontreatment groups that have the most similar covariables [87]. Therefore, referring to previous studies, this paper selects 80 unregenerated projects and calculates the propensity score of unregenerated and regenerated projects by a logit regression model as follows [21,47].

$$\text{Probability of Regeneration} = f\left(\alpha_0 + \gamma_j * \text{Project}_j + \theta_j * \text{Location}_j + \varepsilon_j\right) \quad \text{(A2)}$$

where $Project_j$ is a vector of projects' observable characteristics, including the age and number of buildings, while $Location_j$ is a vector of spatial features, including the distances to the nearest subway station, commercial area, and college. According to the PSM, every urban regeneration project has been matched to an unregenerated project according to the similarity of variables. Then, this paper compares the housing price changes of these projects before and after the period of urban regeneration. If there is no significant disparity between the result with and without the treatment of PSM, the benchmark regression results can be considered robust.

Table A7 reports the results of PSM. It can be seen that the bias between unmatched groups and matched groups is generally reduced. It is acceptable for the bias of most covariables lower than 10%. The treatment effects are estimated based on Equation (1). As shown in Table A8, the final coefficient of $DD$ is $-0.181$, which is significant. To conclude, all the above tests are passed, and the benchmark regression results are confirmed.

**Table A7.** The bias of covariates before and after PSM treatment.

| Variable | Unmatched Matched | Mean | | %Reduct | | t-Test | |
|---|---|---|---|---|---|---|---|
| | | Treated | Control | %Bias | Bias | t | p > t |
| Age | U | 0.409 | 0.274 | 28.300 | 1.170 | 0.245 | . |
| | M | 0.409 | 0.364 | 9.500 | 66.300 | 0.300 | 0.764 |
| Buildings | U | 14.727 | 23.887 | −29.100 | −0.990 | 0.327 | 0.080 * |
| | M | 14.727 | 16.500 | −5.600 | 80.600 | −0.360 | 0.717 |
| DISSS | U | 535.600 | 1721.900 | −47.200 | −1.570 | 0.121 | 0.010 * |
| | M | 535.600 | 589.910 | −3.500 | 92.600 | −0.630 | 0.532 |
| DISC | U | 207.340 | 943.560 | −75.900 | −2.510 | 0.014 | 0.010 * |
| | M | 207.340 | 278.070 | −7.300 | 90.400 | −1.630 | 0.111 |
| DISE | U | 807.740 | 3003.200 | −111.700 | −3.850 | 0.000 | 0.140 * |
| | M | 807.740 | 1651.400 | −42.900 | 61.600 | −2.600 | 0.013 |

**Note:** * means significant at the level of 10%.

**Table A8.** The regression results after PSM.

| | (1) | (2) | (3) | (4) |
|---|---|---|---|---|
| DD | −0.098 | 0.036 | −0.278 *** | −0.181 *** |
| | (0.106) | (0.101) | (0.043) | (0.041) |
| POST | 1.483 *** | 1.658 *** | 0.001 | −0.041 |
| | (0.056) | (0.056) | (0.032) | (0.030) |
| TREATED | −0.927 *** | −0.813 *** | −0.652 | −1.561 *** |
| | (0.075) | (0.077) | (0.688) | (0.206) |
| Area | | | −0.002 | −0.004 |
| | | | (0.001) | (0.002) |
| Dec | | | −0.020 | 0.687 *** |
| | | | (0.043) | (0.022) |
| Bed | | | 0.481 *** | 0.008 |
| | | | (0.059) | (0.076) |

**Table A8.** *Cont.*

|  | (1) | (2) | (3) | (4) |
|---|---|---|---|---|
| Ele |  | 2.534 *** | | 0.337 *** |
|  |  | (0.079) | | (0.093) |
| Age |  | −0.030 *** | | −0.045 *** |
|  |  | (0.004) | | (0.004) |
| Pr |  | 0.017 | | 3.039 ** |
|  |  | (0.016) | | (1.358) |
| Gr |  | 3.775 *** | | −5.775 |
|  |  | (0.351) | | (33.150) |
| Dissub |  | −0.371 *** | | −1.274 *** |
|  |  | (0.059) | | (0.201) |
| Dise |  | 0.176 *** | | −0.526 * |
|  |  | (0.016) | | (0.316) |
| Disc |  | −0.244 *** | | −0.204 |
|  |  | (0.041) | | (0.913) |
| Dispro |  | −0.070 *** | | −0.148 |
|  |  | (0.010) | | (0.235) |
| House FE | No | No | Yes | Yes |
| Time FE | No | No | Yes | Yes |
| Control Variables | No | Yes | No | Yes |
| R-squared | 0.056 | 0.138 | 0.912 | 0.923 |
| Observations | 21316 | 21316 | 21315 | 21316 |

**Note:** *, **, and *** mean significant at the level of 10%, 5%, and 1%, respectively. House FE and Time FE indicate the housing and time fixed effect. Yes and No indicate the variables are controlled or not.

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
