# Peer review of "Positive or Negative: The Heterogeneities in the Effects of Urban Regeneration on Surrounding Economic Vitality—From the Perspective of Housing Price"

_land, doi:10.3390/land13050652_

Round 1

Reviewer 1 Report

Comments and Suggestions for Authors

The problem statement of the study is not well developed in the Introduction. The objective is clear, but the specific research questions are missing. 

Due to the lack of research questions, the focus of the study is not clear. It must be clarified whether economic vitality or housing prices is in the focus of the study? 

In the theoretical section, the treatment of the literature cited is good. However, there is no review of the international literature on rehabilitation, revitalisation and redevelopment (what exactly do these terms mean at international level)? The authors only provide one-sentence definitions from the Chinese literature, which is not enough to understand these processes.

The main shortcoming of the theoretical part is that the authors do not address and discuss the concept of economic vitality. In my opinion, the added value of the study is much more significant in the topic of housing prices, and I therefore recommend that the authors delete the concept and scope of economic vitality from the title and the content of the study. Less will be more in this case.

The content of Table 1 in the methodology is not clear (what do the values in the investment and value columns mean?)

It is not clear how the content of Table 2 relates to the topic.

The mathematical and statistical analyses introduced in the methodological section and used in the presentation of results are of a high quality, but in places unnecessarily over-complicated, which does not get closer to the main message of the study. The results on economic vitality and housing prices are mixed in the study, so the internal coherence of the paper is not as high as expected. If authors decide to retain both topics (i.e. economic vitality and housing prices), they need to be more closely linked in the paper.

The content of Table 3 is not understandable, as the study lacks an explanation of the headings.

The research findings are exciting and novel, however, the results need more detailed professional explanations and it would be essential to compare them with international (mainly North American and European) findings.

Answers to the research questions are missing in the Conclusions.

To sum up, key issues to be addressed are as follows: formulating research questions, clarifying the focus of the study, expanding the theoretical section, placing the research results in an international context in the discussion and answering research questions in the Conclusions.

Comments on the Quality of English Language

The quality of English is good.

Reviewer 2 Report

Comments and Suggestions for Authors

Dear authors,

Congratulations for your interesting research. I have some suggestions on how to make your text more attractive for wider audience.

Your discussion on real estate projects is coherent, what I miss, however, is Life Cycle cost analysis approach, either for historic buildings (see here: DOI: 10.52950/ES.2022.11.2.009) and for new construction (see here: DOI 10.3390/buildings11110524). Including the influence on public tenders may provide an interesting aspect to the readers (see here: DOI: 10.52950/ES.2023.12.1.006).

The context of your study is high regional heterogenity, a resonant issue and task for governments all over the world. I believe that this context should be more deeply developed in order to give your topic a wider and international context. It has been found that less developed regions suffer from deeper magnitude of real estate price cycle and slower overal economic recovery (see here: https://doi.org/10.3390/buildings13040896. Private investments in less developer regions are thus more risky signalling a poverty vicious cycle.

These are interesting points of view to be included and referenced in your research in order to attract more attention from international audience

Good Luck!

Reviewer 3 Report

Comments and Suggestions for Authors

This article mainly discusses the role and impact of urban renewal on housing prices. The article has a clear structure, reasonable methods, and sufficient data to effectively support the proposed hypothesis.

But I still identified a few minor issues.

1. The compass and scale bar are missing from the map in Figure 1.

2. In Table 1, the "Investment" column lacks units. Combining Table 1 and Figure 1, since the author's data can distinguish different types of regeneration, and also conducted discussions for different types, it is recommended that different categories of symbols be used to display the three kinds of regeneration in Figure 1.

3. The "neighborhood" column in Table 2, does the author mean "gated community"?

4. Page 14, Line 430, "Error! Reference source not found".

5. Urban regeneration mostly occurs in urban centers or inner cities, and is highly related to various distance-related factors. Two articles are recommended for the author’s reference:

Wang, D., Li, V. J., & Yu, H. (2020). Mass appraisal modeling of real estate in urban centers by geographically and temporally weighted regression: A case study of Beijing’s core area. Land9(5), 143.

Alves, Sónia, et al. "Urban Regeneration, Rent Regulation and the Private Rental Sector in Portugal: A Case Study on Inner-City Lisbon’s Social Sustainability." Land 12.8 (2023): 1644.

Comments on the Quality of English Language

Minor editing of English language required

Round 2

Reviewer 2 Report

Comments and Suggestions for Authors

I recommend this paper for publication. It presents an interesting research.

Reviewer 3 Report

Comments and Suggestions for Authors

The author has responded to all comments and made appropriate modifications, and I believe that the revised manuscript is suitable for publication.

Comments on the Quality of English Language

Minor editing of English language required